# *Broussonetia papyrifera* Root Bark Extract Exhibits Anti-inflammatory Effects on Adipose Tissue and Improves Insulin Sensitivity Potentially Via AMPK Activation

**DOI:** 10.3390/nu12030773

**Published:** 2020-03-14

**Authors:** Jae Min Lee, Sun Sil Choi, Mi Hyeon Park, Hyunduk Jang, Yo Han Lee, Keon Woo Khim, Sei Ryang Oh, Jiyoung Park, Hyung Won Ryu, Jang Hyun Choi

**Affiliations:** 1Department of Biological Sciences, Ulsan National Institute of Science and Technology (UNIST), Ulsan 689-798, Korea; jssj0612@unist.ac.kr (J.M.L.); ssili77@unist.ac.kr (S.S.C.); Yohan010@unist.ac.kr (Y.H.L.); kkwzang@unist.ac.kr (K.W.K.); jpark@unist.ac.kr (J.P.); 2Natural Medicine Research Center, Korea Research Institute of Bioscience and Biotechnology, Cheong-ju si, Chungcheongbuk−do 28116, Korea; mihyeon@kribb.re.kr (M.H.P.); seiryang@kribb.re.kr (S.R.O.); 3Department of Internal Medicine, Seoul National University, Seoul 110-744, Korea; wony7458@gmail.com

**Keywords:** *Broussonetia papyrifera* root bark, adipocyte, inflammation, AMPK, insulin sensitivity

## Abstract

The chronic low-grade inflammation in adipose tissue plays a causal role in obesity-induced insulin resistance and its associated pathophysiological consequences. In this study, we investigated the effects of extracts of *Broussonetia papyrifera* root bark (PRE) and its bioactive components on inflammation and insulin sensitivity. PRE inhibited TNF-α-induced NF-κB transcriptional activity in the NF-κB luciferase assay and pro-inflammatory genes’ expression by blocking phosphorylation of IκB and NF-κB in 3T3-L1 adipocytes, which were mediated by activating AMPK. Ten-week-high fat diet (HFD)-fed C57BL6 male mice treated with PRE had improved glucose intolerance and decreased inflammation in adipose tissue, as indicated by reductions in NF-κB phosphorylation and pro-inflammatory genes’ expression. Furthermore, PRE activated AMP-activated protein kinase (AMPK) and reduced lipogenic genes’ expression in both adipose tissue and liver. Finally, we identified broussoflavonol B (BF) and kazinol J (KJ) as bioactive constituents to suppress pro-inflammatory responses via activating AMPK in 3T3-L1 adipocytes. Taken together, these results indicate the therapeutic potential of PRE, especially BF or KJ, in metabolic diseases such as obesity and type 2 diabetes.

## 1. Introduction

Inflammation is a protective response against infection, tissue stress, and injury in any tissue and defends and restores physiological functions. However, dysregulated inflammatory processes result in chronic inflammation, which is increasingly seen as a major driver of numerous diseases such as obesity and type 2 diabetes [1]. Obese adipose tissue produces inflammatory cytokines, including tumor necrosis factor (TNF)-α, monocyte chemokine protein (MCP)-1, and interleukin (IL)-6 [1]. Subsequently, the elevated inflammatory stimuli induce the activation of the inhibitor of κB (IκB) kinase (IKK)/NF-κB and c-Jun N-terminal kinase (JNK) pathways, which negatively regulate insulin action in not only adipose tissue, but also other peripheral tissues, such as liver [2]. Thus, the accumulation of pro-inflammatory responses in adipose tissue may be one of the causal factors for insulin resistance. A previous study has demonstrated that pro-inflammatory gene expression is elevated in adipose tissue in the early onset of obesity, but in other tissues, such as liver and skeletal muscle, there is no differences in the expression of inflammatory gene expressions [3]. Thus, adipose tissues appear to act as priming tissues that respond to a high-fat diet (HFD) and initiate inflammation in obesity. Therefore, understanding the inflammatory responses in adipose tissues of obese individuals is of clinical importance.

It has been well demonstrated that AMP-activated protein kinase (AMPK) is a master regulator for energy sensing, which responds to control energy homeostasis. AMPK can be activated by various conditions. Starvation, hypoxia, exercise, and oxidative damages are the main cellular stresses for activating AMPK [4]. There are two well-known upstream kinases: liver kinase B1 (LKB1) and Ca^2+^/calmodulin-dependent protein kinase kinase (CaMKK) can activate AMPK via phosphorylation. Several reports clearly demonstrated that one of the major roles of AMPK is regulating metabolic requirement. For example, AMPK stimulates energy production pathways through fatty acid oxidation, mitochondrial biogenesis, and glucose catabolism. On the other hand, it inhibits energy-consuming pathways, including fatty acids’ synthesis and amino acids’ biogenesis [5]. Thus, dysfunctions of AMPK or downstream signaling pathways could result in metabolic diseases, such as obesity and type 2 diabetes [4].

Interestingly, it has been reported that AMPK could suppress the NF-κB transcriptional activity [6]. The activation of AMPK by AICAR (5-aminoimidazole-4-carboxamide-1-β-D-ribofuranoside) can inhibit colitis [7], autoimmune encephalomyelitis [8], and inflammation after lung injury [9]. In contrast, disrupting AMPK-mediated signaling in hematopoietic-derived cells induced the infiltration of adipose tissue macrophages (ATMs) and hepatic steatosis [10]. In addition, pro-inflammatory responses inhibited the activation of AMPK in adipose tissue and induced the expression of pro-inflammatory genes in vivo [11]. It has been reported that the infiltration of ATMs is significantly increased in AMPKα1^−/−^ mice, and these mice showed increased expression of pro-inflammatory genes, such as IL-6 or TNF-α, in adipose tissue [12]. Together, these observations that AMPK can suppress inflammation have a significant impact on obese-mediated inflammation in adipose tissue.

Paper mulberry (Broussonetia papyrifera) is a deciduous tree that is distributed throughout Asia, and its barks, roots, and fruits are used in traditional Chinese medicine. It has been shown that Broussonetia papyrifera has anti-tyrosinase and antioxidant activity [13,14] and anti-inflammatory activities in cells [15]. Constituents of the roots of this plant, broussochalcone A, kazinol A, and kazinol I, have been reported as inhibitors of lipopolysaccharide-induced nitric oxide (NO) production by suppressing NF-κB activation in macrophages [16,17]. Moreover, kazinol B, a *B. papyrifera*-derived prenylated flavan, has been shown to inhibit NO production [16]. Interestingly, kazinol B enhances glucose uptake via Akt (a serine/threonine kinase) and AMPK activation in adipocytes [17]. Collectively, these reports suggest that *B. papyrifera* might ameliorate inflammation, but to what degree it elicits systemic insulin sensitivity, and by what mechanism, remains unclear. In the present study, we aimed to demonstrate that roots of *B. papyrifera* improve pre-established insulin resistance and identify major bioactive compounds that modulate obese-associated inflammation in adipose tissue.

## 2. Materials and Methods

### 2.1. Plant Material

The root bark of *B. papyrifera* was sampled at Mugo-ri, Gonyang-myeon, Sacheon-si, Gyeongsangnam-do, South Korea, in June 23, 2015 (by Dr. Jin-Hyub Paik). The collected raw materials were deposited in the Korea Research Institute of Bioscience and Biotechnology (KRIBB) and the International Biological Material Center (IBMRC) (KRIBB 0059119) [18]. Of the collected roots, only barks were used for obtaining a better yield.

### 2.2. Preparation of B. papyrifera Root Bark

The target compounds were isolated from dried root bark of *B. papyrifera* as previously described [18]. Briefly, the total *B. papyrifera* root bark extracts (TPRE, yield 10.05%) were separated by SPOT-II MPLC (medium-pressure liquid chromatography) (Gilson, Middleton, WI, USA) using reversed-phase silica gel (YMC-Pack ODS-AQ HG, 20 × 250 mm, 10 μm, Kyoto, Japan) eluted with MeOH-H_2_O to give TPRE Nos. 1–8 (Appendix A and Figure 1B). Broussoflavonol B and kazinol J were found to be components of TPRE No. 6 (PRE) based on the UPLC-PDA-QTof-MS chromatograms (Appendix A). Among these fractions, TPRE No. 6 (PRE) was subjected to a GX-271 (automated liquid handler) semipreparative HPLC system (Gilson, Middleton, WI) using a reversed-phase column (YMC-Pack ODS-AQ-HG, 10μm) and eluted with the MeOH-H_2_O gradient system (55% → 100% MeOH, 70 min) by 25 repeated injections of the samples (2 g/mL methanol dilutions) to yield four fractions (PRE Nos. 6–1~6–4). A further preparation.-HPLC procedure was repeated several times using each condition (see below). The fraction 6–3–5 (345.1 mg) enriched with broussoflavonol B was further isolated by prep.-HPLC (Gilson, Middleton, MA, USA). Broussoflavonol B (44.9 mg) and broussonol D (29.4 mg) were isolated using a YMC-Pack pro C8 column. The fraction 6–3–7 enriched with kazinol J was further isolated by prep.-HPLC (PLC 2020) with a gradient system of MeOH-H_2_O, and (-)–(2*S*)–kazinol I (7.6 mg), kazinol J (48.7 mg), broussoflavonol C (215.7 mg), and broussonol G (40.6 mg) were obtained. The isolated compounds were purified as described previously, and purity the was more than 95.0%, as determined by ultra-performance liquid chromatography [18]. Two compounds were characterized using spectroscopic data, including ^1^H, ^13^C NMR, and HRMS, in comparison with previously published data [16,19]. An ACQUITY UPLC™ system (Waters Corporation, Milford, MA, USA) equipped with a binary solvent delivery manager and a photodiode array (PDA) was used for UPLC (ultra-performance liquid chromatography) analysis. HRMS analysis was performed using an ultra-performance liquid chromatography quadrupole time of flight mass spectrometry (UPLC-QTOF-MS) equipped with an electrospray ionization (ESI) interface (Waters Q-TOF PremierTM, Waters Corporation, Milford, MA, USA). The NMR analysis was carried out using a Fourier transform (FT)-NMR spectrometer (JEOL ECZ500R, JEOL Ltd., Akishima, Tokyo, Japan) for 1D spectra (^1^H NMR and ^13^C NMR). The overall processes are described in Appendix A. All extracts and single compounds for the experiments were prepared by dissolving in dimethyl sulfoxide (DMSO, Sigma, St. Louis, MO, USA).

### 2.3. Cell Culture and Adipocyte Differentiation

3T3-L1, Raw264.7, and HEK293 cells were obtained from the American Tissue Culture Collection (ATCC, Manassas, VA, USA) and cultured in Dulbecco’s Modified Eagle’s Medium (DMEM, Life Technologies, NY, USA) with 10% bovine calf serum (Invitrogen, Gaithersburg, CA, USA) and 10% fetal bovine serum (Atlas, CO, USA), respectively. Adipocyte differentiation was induced by treating cells with DMEM containing 10% FBS, 0.5 mM isobutylmethylxanthine (IBMX), 1 μM dexamethasone, and 850 nM insulin. After 48 h, the medium was replaced every other day with DMEM containing 10% FBS and 850 nM insulin. All chemicals for cell culture were obtained from Sigma-Aldrich (St. Louis, MO, USA) unless otherwise indicated. After 6–7 days from initiation of differentiation (Appendix A), we treated compounds as indicated concentrations and time. After treating fully differentiated adipocytes with PRE, BF, or KJ for 24h, cell morphology was monitored by an inverted microscope (ZEISS, Oberkochen, Germany) (Appendix A). For Oil Red O staining, 3T3L1 preadipocytes or differentiated 3T3L1 adipocytes treated with/without PRE, BF, or KJ as indicated concentrations were stained with the Oil Red O staining kit according to the manufacturer´s recommendations (Biovision, Inc., Milpitas, CA, USA).

### 2.4. Reporter Gene Assay

HEK-293 cells were transfected with NF-κB-responsive luciferase reporter (Promega, San Luis Obispo, WI, USA) and pRL-Renilla using Lipofectamine 2000 (Invitrogen, Gaithersburg, CA, USA). Following an overnight transfection, the cells were treated with PRE, BF, or KJ for 24 h, followed by treatment with TNF-α (10 ng/mL) for 6 h. DMSO was used as the vehicle. The cells were harvested, and reporter gene assays were carried out using the Dual-Luciferase kit (Promega, San Luis Obispo, WI, USA). Luciferase activity was normalized to Renilla activity.

### 2.5. Nitric Oxide Production and Cell Viability in Raw 264 Cells

Raw 264 macrophages were seeded in 96 well plates (2.0 × 10^5^ cells/mL) and treated with PRE, BR or KJ as the indicated concentration with/without lipopolysaccharide (LPS, Sigma, St. Louis, MO, USA) for 24 h. For NO production, the amount of NO was calculated by measured nitrate in media using Griess reagent according to the manufacturer´s recommendations (Sigma, St. Louis, MO, USA). For cell viability, cells were determined using the MTT solution (3-(4,5-dimethylthiazol-2-yl)-2,5-diphenyltetrazolium bromide), Sigma, St. Louis, MO, USA) at 0.5 mg/mL. The purple formazan crystals were dissolved in DMSO, and the absorbance was recorded on a microplate reader at a wavelength of 570 nm.

### 2.6. Animal Experiment

All animal experiments were performed according to the procedures approved by Ulsan National Institute of Science and Technology’s Institutional Animal Care and Use Committee (UNISTIACUC-19-04). Seven-week-old male C57BL/6J mice (DBL, samsung, Korea) were fed a high fat diet (60% kcal fat, D12492, Research Diets Inc., New Brunswick, NJ, USA) for 10 weeks The mice were housed (*n* = 4/cage) and granted free access to food and water. Food and water were changed once a week. For glucose tolerance tests (GTT), mice were intraperitoneally (i.p.) injected daily with 40 mg/kg of PRE or vehicle (saline containing 5% DMSO and 5% Tween 80 (Sigma, St. Louis, MO, USA) for 7 days and fasted for 16 h (6 p.m. to 10 a.m.) prior to i.p. injection of D-glucose (2 g/kg body weight). PRE solution prepared at 4 mg/mL in saline containing 5% DMSO and 5% Tween 80 was injected in an amount of 10 μL/g of body weight. Fasting insulin was determined using the ultrasensitive mouse insulin ELISA kit (Crystal Chem., Eik Grove Village, IL, USA). Once mice were sacrificed, isolated adipose tissue and liver were weighed and immediately frozen in liquid nitrogen and then used for Western blot analysis and gene expression analysis.

### 2.7. Histological Analysis

Liver sections were embedded in paraffin and stained with hematoxylin and eosin (H&E) to visualize hepatocytes and lipid droplets in the tissues. Sections were analyzed by an inverted microscope (ZEISS, Oberkochen, Germany).

### 2.8. Western Blot Analysis

Each sample (cells or tissues) was lysed with RIPA lysis buffer containing protease and phosphatase inhibitor (Sigma-Aldrich, St. Louis, MO, USA). An equal amount of protein was separated on SDS-PAGE and transferred onto nitrocellulose membranes (GE Healthcare, Chigago, IL, USA). The membranes were blocked in a 5% bovine serum albumin (BSA) blocking buffer and incubated with specific primary antibodies for phospho-NF-κB, NF-κB, phospho-IκB, IκB, phospho-AMPK, AMPK, phospho-ACC, and ACC (Cell signal technology, Danver, MA, USA) at 4 °C overnight. The signals were detected using an ECL detection kit (GE Healthcare, Chigago, IL, USA), followed by incubation with horseradish peroxidase-conjugated secondary antibodies (Thermofisher, Gaithersburg, CA, USA). We quantified the band intensity by using the ImageJ program (NIH, Bethesda, MD, USA).

### 2.9. Gene Expression

Total RNA was isolated from cells or tissues using TRIzol reagents (Invitrogen, city, CA, USA). The RNA was reverse-transcribed using the ABI reverse transcription kit. Quantitative PCR reactions were performed with SYBR green fluorescent dye using an ABI9300 PCR machine. Relative mRNA expression was determined by the ∆∆-Ct method normalized to TATA-binding protein (TBP) levels.

### 2.10. Statistical Analysis

Data were presented as the means + S.E.M. Statistical significance was estimated by an unpaired *t*-test for comparisons between two conditions. A one-way ANOVA was used for comparisons between more than two conditions. Dunnett’s post hoc test was used for multiple comparisons. All statistics were performed with GraphPad Prism 7.0 software (GraphPad, San Diego, CA, USA).

## 3. Results

### 3.1. PRE Suppresses TNF-α-Induced NF-κB Activity

To investigate the effects of PRE on inflammation, we first tested NF-κB transcriptional activity of TPRE because NF-κB is an essential regulator of pro-inflammatory response (Appendix A). As shown in Figure 1A, TPRE suppressed NF-κB transcriptional activity induced by TNF-α, and it was dose-dependent. To further evaluate the effect of TPRE for inhibiting NF-κB activity, we partitioned TPRE using medium-pressure liquid chromatography into eight sub-fractions (Appendix A), which were then used to assay NF-κB transcriptional activity. Of the eight sub-fractions, four fractions significantly inhibited NF-κB transcriptional activity induced by TNF-α. Sub-fraction 6 (PRE) had the most potent activity for inhibiting NF-κB activation and dose-dependently repressed NF-κB transcriptional activity (Figure 1B,C). 5-Aminoimidazole-4-carboxamide ribonucleotide (AICAR), an AMP analog, was used as the positive control for AMPK activation.

### 3.2. PRE Suppresses Pro-Inflammatory Gene Expression in Adipocytes

Next, we further tested the effect of PRE on inflammatory responses. Differentiated adipocytes were treated with TNF-α, and we examined the effects of PRE on the expression of pro-inflammatory genes. As shown in Figure 2A, PRE repressed the TNF-α-mediated pro-inflammatory gene, and this effect was dose-dependent. Furthermore, PRE blocked lipopolysaccharide (LPS)-induced pro-inflammatory response in Raw264.7 cells, as previously reported (Appendix A) [18]. AMPK is a well-known, important inflammatory suppressor, and AMPK signaling critically regulates inflammation in many cell types [6]. Therefore, to further investigate the molecular mechanism involved in PRE-associated repression of the NF-κB signaling pathway, we first tested whether PRE activated AMPK. In 3T3-L1 adipocytes, PRE enhanced the phosphorylation of AMPK, while PRE did not affect AMPK protein level. This phosphorylation was blocked by compound C, a specific inhibitor of AMPK (Figure 2B). Similarly, the phosphorylation of acetyl-CoA carboxylase (ACC), a substrate of AMPK, was enhanced by PRE treatment, and its phosphorylation was blocked by compound C treatment. PRE treatment decreased TNF-α-induced phosphorylation of IκB and NF-κB in adipocytes (Figure 2C). In addition, pretreatment with AICAR specifically inhibited TNF-α-mediated pro-inflammatory signaling. Pretreatment with compound C blocked PRE-induced suppression of phosphorylation of IκB and NF-κB in 3T3-L1 adipocytes (Figure 2C). Together, these results strongly suggest that PRE suppresses TNF-α-mediated pro-inflammatory gene expression by activating AMPK.

### 3.3. PRE Improves Obesity-Associated Systemic Glucose Tolerance

Next, we determined whether PRE exhibited anti-diabetic activities in vivo. Ten-week-HFD-fed C57BL/6 mice were used for a glucose tolerance test (GTT). PRE and glucose were administrated intraperitoneally, because the components of PRE have been reported to inhibit alpha-glucosidase to exclude the possibility that oral glucose and PRE administration suppress glucose absorption from the gut [20]. The PRE-treated group showed increased glucose tolerance compared to that in control mice (Figure 3A). Plasma glucose levels in PRE-treated mice, as determined by the area under the curve (AUC), were significantly suppressed compared to those of control mice. Furthermore, the fasting insulin level showed a tendency to decrease (Figure 3C), suggesting that PRE ameliorates obesity-induced glucose intolerance. Body weight (Figure 3B), adipose tissue weight (Appendix A), and liver weight (Appendix A) were not changed by treatment with PRE.

### 3.4. PRE Ameliorates Adipose Tissue Inflammation

Our initial results showed that PRE activated AMPK and suppressed pro-inflammatory mediators in vitro. Therefore, we tested whether PRE activated AMPK and NF-κB in obese adipose tissue in vivo. Upon treatment with PRE, NF-κB phosphorylation was significantly decreased, whereas AMPK phosphorylation was increased in epididymal white adipose tissue (eWAT) (Figure 4A). Next, we assessed the effects of PRE on inflammation in adipose tissue. As shown in Figure 4B, the expression of pro-inflammatory genes (Il-1β and inducible nitric oxide synthase (iNOS)) was significantly reduced in eWAT after treatment with PRE. However, marker genes of macrophages (*F4/80*, *cd11b*, and *cd68*) or marker genes of the M2 macrophage, including arginase-1 (*Arg-1*), mannose receptor C-type 1 (*Mrc-1*), macrophage galactose binding lectin (*Mgl*), and *Ym-1* (chitinase-like 3) were not changed (Appendix A). Interestingly, PRE-treated HFD-fed mice showed significantly decreased expression of lipogenic genes (fatty acid synthase (*Fasn*), sterol regulatory element-binding protein 1 (*Srebp-1c*), and *Acc-1*) in adipose tissue (Figure 4C). These results indicated that PRE had anti-inflammatory and anti-lipogenic activities in adipose tissue.

### 3.5. PRE Ameliorates Hepatic Steatosis

Next, we examined whether PRE prevented hepatic steatosis, which is increased by obesity. Histological observations revealed that PRE significantly suppressed hepatic steatosis in obese mice induced by HFD (Figure 5A and Appendix A). Because AMPK plays crucial roles in suppressing hepatic steatosis [4] and PRE activated AMPK in adipose tissue, we examined AMPK signaling in liver. As shown in Figure 5B, treatment with PRE increased AMPK phosphorylation in liver. Furthermore, PRE significantly decreased lipogenic gene expression (*Srebp-1c* and stearoyl-CoA desaturase-1 (*Scd-1*)) (Figure 5C). In addition, PRE increased the expression of acyl-CoA synthetase long-chain (*Acsl*), very-long-chain acyl-CoA dehydrogenase (*Vlcad*), and short-chain acyl-CoA dehydrogenase (*Scad*), which are involved in fatty acid oxidation (Figure 5D). Taken together, these results suggested that PRE activated AMPK, which improved fatty liver.

### 3.6. Broussoflavonol B and Kazinol J are Bioactive Compounds of PRE

To identify the bioactive compounds in PRE that activate AMPK, we isolated 20 compounds via methanolic extraction (Appendix A). Among them, we found that broussoflavonol B (BF) and kazinol J (KJ) dramatically increased AMPK phosphorylation in 3T3-L1 adipocytes (Figure 6A). Both BF and KJ increased AMPK and ACC phosphorylation, and compound C significantly blocked them (Figure 6B). Furthermore, BF and KJ significantly suppressed TNF-α-induced NF-κB transcriptional activity (Figure 6C). Consistent with NF-κB activity, treatment with BF and KJ downregulated TNF-α-stimulated pro-inflammatory gene expression (*Il-6*, *Mcp-1*, and *iNOS-*only in the BF-treated group) in adipocytes (Figure 6D). In addition, both BF and KJ decreased IκB degradation and NF-κB phosphorylation, and compound C significantly blocked them (Figure 6E). They also suppressed LPS-mediated NO production in Raw264.7 (Appendix A). Together, these results strongly indicated that BF and KJ were bioactive compounds of PRE and could block an inflammatory response through blocking the NF-κB signaling pathway via AMPK activation.

## 4. Discussion

Obesity is very closely related to chronic and low-grade inflammation. Many reports have suggested that insulin resistance accompanies chronic inflammation and abnormal mediator secretion in obese adipose tissue [21], indicating that reduction of tissue inflammation could be an indispensable target for improving obesity-related metabolic syndromes. It has been reported that roots of *B. papyrifera* have been used as a suppressant for edema in traditional Chinese medicine [22]. The core phytochemicals in the roots of *B. papyrifera* are flavonols, flavans, and chalcones [23]. There is emerging evidence that these core phytochemicals have anti-inflammatory activities. [15,16,24]. In the present study, we focused on the anti-inflammatory effects of PRE in adipose tissue and investigated whether it could ameliorate systemic insulin resistance in obese mice. PRE potently inhibited TNF-α-induced inflammatory responses by suppressing NF-κB activation in adipocytes. This effect improved not only glucose tolerance, but also hepatic steatosis in HFD-induced obese mice. Using the FDA guideline to calculate human equivalent doses (HED), the HED of this effective dose of PRE (40 mg/kg) was 195 mg/day in humans. This dose was lower than that of metformin (850 mg~2550mg/day) even though PRE was a crude extract. Furthermore, BF and KJ, the isolated phytochemicals from PRE, were the bioactive compounds that suppressed inflammatory responses in 3T3-L1 adipocytes through blocking NF-κB signaling. Of significance, these effects were partially dependent on AMPK activation in adipocytes (Appendix A).

AMPK has been shown to have strong anti-inflammatory activity via inhibiting inflammatory responses in various in vivo models [6]. Various studies have shown that AMPK inhibits pro-inflammatory signaling in many tissues and cells, especially adipocytes and macrophages, which are the main cell types of adipose tissue [25,26]. In addition, AMPK ameliorates insulin resistance in obesity [10,11,25,26]. It has been reported that AMPK activators specifically suppressed the expression of pro-inflammatory genes and the activation of NF-κB-mediated signaling [27,28]. The molecular mechanism for NF-κB activation is: (1) the degradation of IκB in a ubiquitin-dependent manner is triggered through its phosphorylation by IKK; (2) IκB degradation results in nuclear translocation of NF-κB [29]. Based on these results, we proposed the roles of PRE and its constituents to inhibit NF-κB transcriptional activity through the blockade of phosphorylation-mediated IκB degradation and NF-κB phosphorylation; this effect was reversed by inhibition of AMPK. Thus, PRE-mediated NF-κB inactivation would likely be accompanied by inhibition of the TNF-α-stimulated IKK/IκB/NF-κB signaling pathway. In addition to the identified signaling pathways, it remains to be elucidated whether other molecular mechanisms for the AMPK-mediated anti-inflammatory effect of PRE exist. AMPK has several phosphorylation targets [30], but there are some reports suggesting that AMPK could inhibit NF-κB activity and its signaling indirectly via sirtuin 1 (SIRT1) [11], the forkhead box O (FoxO) family [30], and peroxisome proliferator-activated receptor γ co-activator (PGC-1α) [31], which are known as downstream mediators of AMPK. These mediators could inactivate the p65 subunit of NF-κB, and the expression of pro-inflammatory genes is subsequently repressed. Thus, further studies related to the underlying mechanism of how PRE and its active components regulate AMPK activation and its resultant anti-inflammatory effect are needed.

Obesity-associated adipocyte dysfunction by chronic inflammation in adipose tissue contributes to developing hepatic steatosis: (1) excessive free fatty acids from adipose tissue, which stimulate inflammation, could be delivered to liver; thus, triglycerides could be accumulated in liver; (2) adipose tissue secretes pro-inflammatory cytokines including TNF-α, which exacerbate inflammation in liver and induce hepatic steatosis [1]. Here, we demonstrated that PRE did not alter eWAT mass and adiposity, by it ameliorated adipose tissue inflammation indicated by significant decreases in the pro-inflammatory signaling and pro-inflammatory genes’ expression. Furthermore, PRE enhanced AMPK activation, which regulates gene expression related to fatty acid oxidation and lipogenesis. Thus, improved adipose tissue inflammation and AMPK activation by PRE could ameliorate hepatic steatosis.

## 5. Conclusions

We demonstrated that PRE and its active components had potential therapeutic effects on ameliorating inflammation in both adipose tissue and liver. Furthermore, PRE could reduce hepatic steatosis and improve glucose homeostasis. In addition, we proposed that the activation of AMPK by PRE and its active components could be the underlying molecular mechanism by which they have anti-inflammatory effects. Taken together, these finding demonstrated the beneficial effects of *B. papyrifera* root and its phytochemicals and indicated their potential as candidates for targeting AMPK for the treatment of obesity and/or type 2 diabetes.

## Figures and Tables

**Figure 1 nutrients-12-00773-f001:**
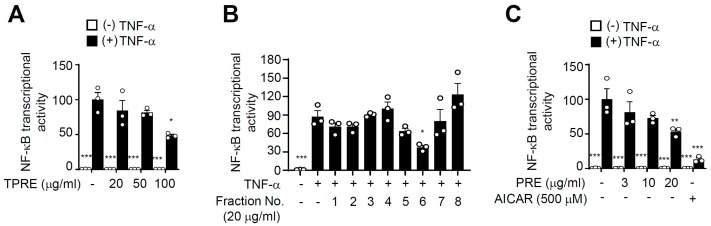
The effects of B. papyrifera root bark extracts on TNF-α-induced NF-κB activity. HEK-293 cells were transfected with the NF-κB-responsive luciferase reporter and pRL-Renilla. The cells were treated with total B. papyrifera (total *B. papyrifera* root bark extracts, TPRE) extracts (**A**), sub-fractions of TPRE (**B**), Sub-fraction No.6 (PRE), and 5-aminoimidazole-4-carboxamide ribonucleotide (AICAR) as the indicated concentration for 24 h (**C**), followed by treatment with TNF-α (10 ng/mL) for an additional 6 h. The cells were harvested, and luciferase activity was measured. Data are shown as the mean ± S.E.M. (*n* = 3) * *p* < 0.05; ** *p* < 0.001; *** *p* < 0.0001 compared to TNF- α-only-treated group.

**Figure 2 nutrients-12-00773-f002:**
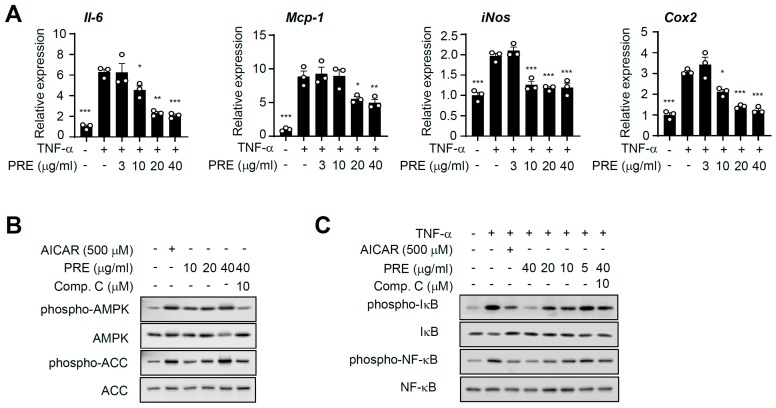
The effects of PRE on TNF-α-induced pro-inflammatory gene expression and AMPK activation in 3T3-L1 adipocytes. (**A**) 3T3-L1 adipocytes were pre-incubated with the indicated concentration of PRE for 24 h, followed by treatment with 20 ng/mL TNF-α for 5 h. Total RNA was isolated, and the mRNA expression level of each gene was analyzed by real time-PCR. Data are shown as the mean ± S.E.M. (*n* = 3) * *p* < 0.05; ** *p* < 0.001; *** *p* < 0.0001 compared to the TNF-α-only-treated group. (**B**) After 3T3-L1 adipocytes were pre-incubation with/without compound C for 1 h, cells were treated with PRE or AICAR for an additional 2 h. The expression of phospho-AMPK, AMPK, phospho-ACC, and ACC was analyzed by Western blotting. (**C**) 3T3-L1 adipocytes were pretreated with the indicated concentration of PRE for 24 h and AICAR for 2 h before 20 ng/mL TNF-α treatment for 30 min. Compound C (Comp. C) was pretreated for 1h before PRE treatment. The expression of phospho-IκB, IκB, phospho-NF-κB, NF-κB, and tubulin was analyzed by Western blotting.

**Figure 3 nutrients-12-00773-f003:**
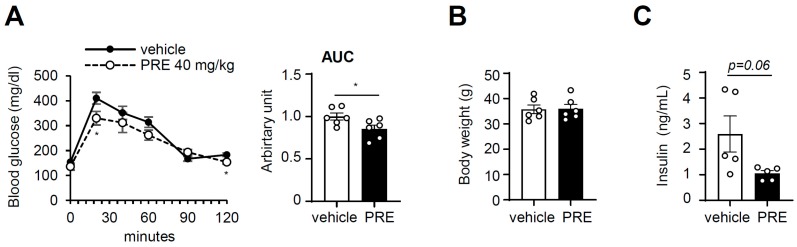
PRE improved glucose tolerance in diet-induced obese (DIO) mice. Seven-week-old male C57BL/6 mice were fed on HFD for 10 weeks, and then PRE was intraperitoneally administrated for a week. (**A**) Intraperitoneal glucose tolerance test (IPGTT) was performed as described in the experimental section. Body weight (**B**) and fasting insulin (**C**) levels were measured at the end of the experiment. Data are shown as the mean ± S.E.M. (*n* = 6) * *p* < 0.05; ** *p* < 0.001; *** *p* < 0.0001 vs. HFD-fed vehicle group; AUC: area under the curve.

**Figure 4 nutrients-12-00773-f004:**
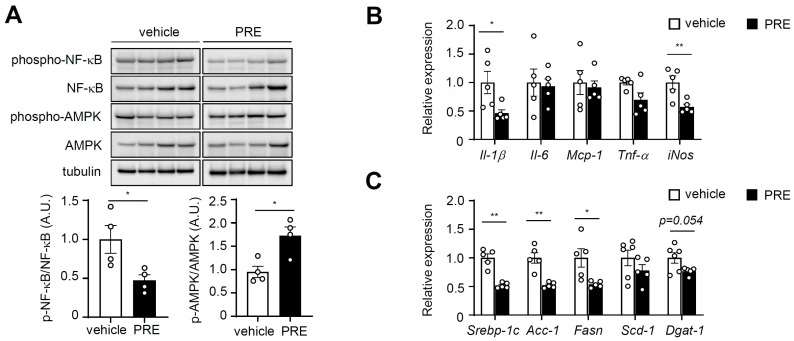
PRE ameliorated adipose tissue inflammation. Adipose tissue was isolated from DIO mice in the non-fasted state. (**A**) The extracts were analyzed with phospho-NF-κB, NF-κB, phospho-AMPK, AMPK, and tubulin by Western blot. A.U., arbitrary units. Data are shown as the mean ± S.E.M. (*n* = 4) * *p* < 0.05; ** *p* < 0.001; *** *p* < 0.0001 vs. HFD-fed vehicle group. The mRNA levels of pro-inflammatory genes (**B**) and lipogenic genes (**C**) were analyzed by quantitative real-time PCR. Data are shown as the mean ± S.E.M. (*n* = 5) * *p* < 0.05; ** *p* < 0.001; *** *p* < 0.0001 vs. HFD-fed vehicle group.

**Figure 5 nutrients-12-00773-f005:**
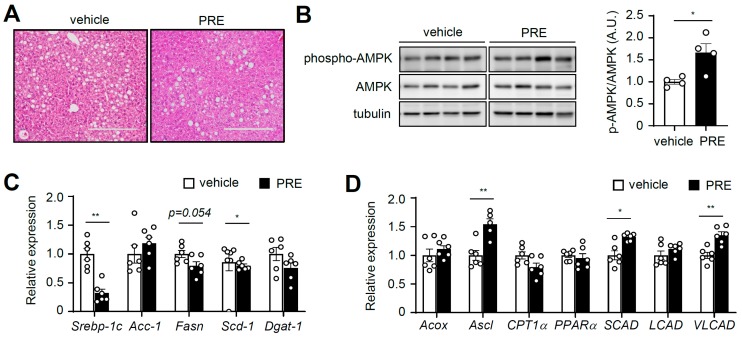
PRE ameliorated hepatic steatosis. Liver was isolated from vehicle- or PRE-treated DIO mice in the non-fasted state. (**A**) Liver sections for H&E staining (the scale bar is 125 μm). (**B**) The extracts were analyzed with phospho-AMPK, AMPK, and tubulin by Western blot. Data are shown as the mean ± S.E.M. (*n* = 4) * *p* < 0.05; ** *p* < 0.001; *** *p* < 0.0001 vs. HFD-fed vehicle group. The mRNA levels of lipogenic genes (**C**) and fatty acid oxidation related genes (**D**) were analyzed by quantitative real-time PCR. Data are shown as the mean ± S.E.M. (*n* = 6) * *p* < 0.05; ** *p* < 0.001; *** *p* < 0.0001 vs. HFD-fed vehicle group.

**Figure 6 nutrients-12-00773-f006:**
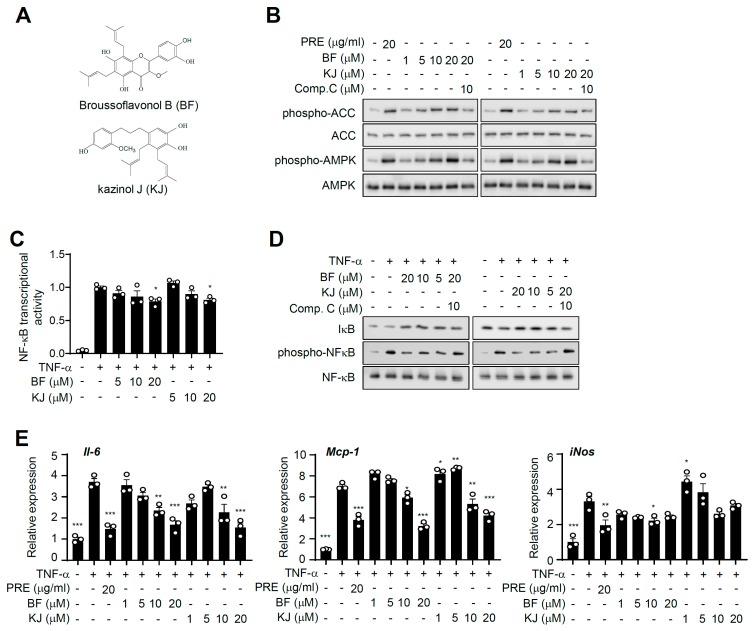
Broussoflavonol B (BF) and kazinol J (KJ) are bioactive compounds of PRE. (**A**) Chemical structure of Broussoflavonol B (top) and kazinol J (bottom). (**B**) 3T3-L1 adipocytes were pre-incubated with/without compound C (10 μM) for 1 h, and then, cells were treated with BF or KJ for an additional 2 h. The expression of phospho-AMPK, AMPK, phospho-ACC, and ACC was analyzed by Western blotting. (**C**) HEK-293 cells were transfected with the NF-κB-responsive luciferase reporter and pRL-Renilla. The cells were treated with BF and KJ as the indicated concentration for 24 h, followed by treatment with TNF-α (10 ng/mL) for an additional 6 h. The cells were harvested, and luciferase activity was measured. Data are shown as the mean ± S.E.M. (*n* = 3) * *p* < 0.05; ** *p* < 0.001; *** *p* < 0.0001 vs. the TNF-α-only-treated group. (**D**) After 3T3-L1 adipocytes were pretreated with/without compound C, cells were incubated with/without BF or KJ for 24 h and stimulated with TNF-α (20 ng/mL) for 30 min. The expressions of phospho-IκB, IκB, phospho-NF-κB, NF-κB, and tubulin were analyzed by Western blotting. (**E**) 3T3-L1 adipocytes were pre-incubated with the indicated concentration of BF or KJ for 24 h, followed by treatment with 20 ng/mL TNF-α for 5 h. Total RNA was isolated, and the mRNA expression level of each gene was analyzed by quantitative real-time PCR. Data are shown as the mean ± S.E.M. (*n* = 3) * *p* < 0.05; ** *p* < 0.001; *** *p* < 0.0001 vs. the TNF-α-only-treated group.

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
