# Peer review of "Broussonetia papyrifera* Root Bark Extract Exhibits Anti-inflammatory Effects on Adipose Tissue and Improves Insulin Sensitivity Potentially Via AMPK Activation"

_nutrients, 2020, doi:10.3390/nu12030773_

Round 1

Reviewer 1 Report

In this report, Lee et al elucidate the mechanistic role and therapeutic potential of Broussonetia papyrifera root bark extract to mitigate cytokine (in vitro) or high fat diet (in vivo) induced inflammation. The study seems well-performed and conceptually interesting, although few concerns arose during rigorous assessment of the submitted report. Please find my suggestion below:

General comments:

  1. While the overall findings from in vitro assays are supported by the in vivo protocol, this reviewer would strongly encourage to a) improve the statistics and b) repeat the animal experiment given the low n-size enhancing the risk of false positive. As all downstream results from this protocol are biologically dependent (arise from the same mice), such false positive results would not be countered by extensive testing.

  1. It is pivotal for interpreting the in vivo data that the authors provide daily food intake during the 7-days treatment as even minor fluctuation between the groups could mediate the reported observations (diminished NALFD, Adipose inflammation, AMPK activation etc.), as they are all highly affected by nutrient influxes.

  1. It would moreover be relevant to relate the used in vivo dose of PRE (40mg/kg BW) to human consumption.

  1. I am surprised to see that the reviewers did not include a LFD reference group in their in vivo setup to corroborate HFD-induced weight gain. The end BW is surprisingly low for this mouse strain and experimental setup.

  1. It would beneficial (read highly encouraged) if the authors would depict their bar plots with individual dots per sample, allowing the reader to visually appreciate data distribution between and within groups). Given the rather low n-size and that the data are unlikely to follow Gaussian distribution, I further suggest to depict all graphs as median ± interquartile range.

  1. As root bark is a food supplement I was surprised to see that the authors chose to administer it intraperitoneally instead of orally. Please provide a sound rationale for this decision – the same goes for the glucose challenge during GTT.

  1. Please discuss whether the anti-inflammatory effects observed are adipocyte-specific, as no effect was seen on typical macrophage effector molecules (in vivo). To substantiate any claims to this end, it might be beneficial to run (at least some) of the in vitro assays on macrophage cell lines in addition to the currently used adipocytes.

  1. I was surprised to see that the authors did not pursue the in vivo efficacy of any of the two identified effecter molecules (BF and KJ)?

  1. On a similar note, the in vitro concentration of PRE is based on mass (Figure 1 and 2), whereas its derivates are depicted in molar concentrations. Please be consistent and use molar throughout, hence providing the reader a change to look at the potency of the different extracts. To this end, it would also be helpful if the authors could educate the reader on the relative concentration of BF and KJ in the intact PRE.

  1. Title: The authors cannot state via AMPK without using relevant KO models… As a minimum, they should add ‘potentially via’. To further substantiate their claims, ot would be relevant with AMPK KD cell experiments to show that AMPK signaling is necessary for the immunometabolic effects of PRE and its derivates (BF an d KJ).

Specific comments:

Introduction:

  1. Line 65-67 needs a bit of grammatical and linguistic help… Unclear what the authors mean. When revising this sentence, please also tone it down to avoid phrases as ‘the strong evidence…’. Suggest to delete ‘strong’ and rephrase to include ‘the observations corroborate’ rather than ‘the evidence’.

  1. Line 76: Delete ‘strongly’.

  1. Line 78: Change ‘tried’ to ‘aimed’ or alternatively delete ‘tried to’.

  1. Line 79: Change ‘the major bioactive…’ to ‘a major bioactive…’

Materials and Methods:

  1. Section 2.4: Did the authors include a mock/vehicle control? It seems so from the result section bu tit is not explicitly mentioned in the M&M. Please amend.

  1. Section 2.5:
    1. How were the animal housed? Singly or grouped?
    2. Overnight fast? Please specify the hours. Was it 12h from e.g. 7PM to 7AM allowing the mice to eat prior fasting or was it from e.g. 4PM to 8AM, which essentially corresponds to 24 hours as mice a nocturnal animals, hence have negligible food-intake throughout the day.
    3. How often did the authors change the food, water etc.? How was the water treated.?
    4. The authors provide the amount of PRE, but not the concentration/volume administered. Please specify.

  1. Section 2.8:
    1. Would suggest to depict individual values (bar plots showing all points) to better appreciate data distribution and allow the reader to instantly assess the number sample number per group for each assay.
    2. Before statistical testing, please assess if the data follows Gaussian distribution and apply the relevant statistical method accordingly.
    3. The authors describe their statistical approach to be students T-test when comparing two groups, but what analysis/post hoc test did they apply to adjust for multiple comparisons when relevant (e.g. Figure 1A-C, 2A, 6C-E)?
    4. It will be important that the authors also correct for multiple testing hypothesis in Figure 4B-C and 5C-D to mitigate the risk of false positive.

Results:

  1. Please explain what AICAR is and why you have used it as a positive controls (Figure 1C). In its current version and without any reference to the compound, the relevance is not readily accessible for the average reader. It is mentioned in the description of Figure 2 but not 1. Please move the initial description up to Figure 1.

  1. Figure 1: The authors mentions that the data (all panels) are representative of three independent experiments, but they do not inform of the actual n-size per experiment. Again, individual dots would help a lot on this matter.

  1. The concentrations (20µg/ml) seems rather high for in vitro assay. It is 2.000 times higher than the TNF-a doses used…

  1. Figure 2C: It would further be appreciated if the PRE concentrations could go from low -> high in panel C as it does in all other panels. Change of orientation does not help on the interpretation.

  1. Figure 3A: I suggest to change the symbols to open circles for the PRE group, allowing the reader to better distinguish between the groups. Also, it is difficult to believe the P values in this panel (especially T120), please double check n-size etc.

  1. Figure 3B: Please provide a weekly bodyweight recording, compare it to LFD fed reference mice and monitor the food intake on a daily basis the 7 days the treatment persist as a modestly reduced food intake could explain the observed in vivo findings.

  1. Figure 3 C: It is not clear whether the insulin levels are in fasted or fed state. Please specify.

  1. I wonder if the authors have insulin data for the entire GTT. Would be beneficial.

  1. Figure 4B and C: As mentioned above, please adjust for multiple comparisons as all analyses in these panels are related, and so some targets may show up as significantly different merely by chance (false positive).

  1. Line 231-234: Again, please tone down some of the statements and delete ‘strongly indicate that PRE has strong anti-inflammatory..’ At best, the results indicate that there are some anti-inflammatory activity. Once adjusting for multiple comparisons, a maximum two target genes will come out as significantly decreased.

  1. Line 251: Delete strongly.

  1. Figure 5A: It would be helpful with an overview picture of the entire liver slide followed by a high resolution (10x or 20x zoom) snap shot. I further suggest to include pictures of all livers in the supplementary file. It is a bit overreaching to conclude diminished steatosis based on a snapshot of a single sample per group. To this end, the manuscript would also benefit from having measured hepatic TG to biochemically support the reported histological findings.

  1. Figure 5C and D: Adjust for multiple comparisons.

Discussion:

  1. Line 295: Change ‘In the present study, we tried to focus…’ to ‘In the present study, we focused’.

  1. Line 301: Based on the provided results the authors cannot state that the effects of PRE, BF and KJ are dependent on AMPK as oppose to cooccur with AMPK activation. Please only phrase it as dependent on, if you have formally shown this, either in vitro or in vivo assays, by relevant KD or KO models.

Author Response

General comments:

  1. While the overall findings from in vitro assays are supported by the in vivo protocol, this reviewer would strongly encourage to a) improve the statistics and b) repeat the animal experiment given the low n-size enhancing the risk of false positive. As all downstream results from this protocol are biologically dependent (arise from the same mice), such false positive results would not be countered by extensive testing.

→ Thank you for your comments. As reviewer’s comment, we have re-analyzed statistics for all data. Statistical significance was estimated by an unpaired t‐test for comparisons between two conditions. A one‐way ANOVA was used for comparisons between more than two conditions. Dunnett’s post hoc test was used for multiple comparisons. All statistics were performed with GraphPad Prism 7.0 software. We have rewritten section 2.8 statistical analysis.

For in vivo experiment, the major revision time of the “Nutrients” journal is limited (within 10 days), we were not able to repeat in vivo experiments. We are very sorry we cannot response to your comment for animal experiment. However, recently, numbers of animal for in vivo experiment have been minimized for animal welfare and safety. In general, 5-10 of animals have been used for experiments and statistical analysis. We performed the experiments with 5-6 mice. Thus, I think it is not too small n-size as you concerned.

  1. It is pivotal for interpreting the in vivo data that the authors provide daily food intake during the 7-days treatment as even minor fluctuation between the groups could mediate the reported observations (diminished NALFD, Adipose inflammation, AMPK activation etc.), as they are all highly affected by nutrient influxes.

è This is a very good point. Unfortunately, we did not measure the food intake when we performed animal experiment. However, the significant difference of the body weight between vehicle and PRE treated groups was not observed during experiment. We started experiment when the average body weight was 41.51g and 41.58g for vehicle and PRE treated group, respectively. These groups were weighted 40.03g and 40.11g in previous day of GTT. The average body weight for vehicle and PRE treated group was 35.75g and 35.4g at the end of the experiment, respectively. Thus, we inferred from body weight that there might be no difference for food intake between two groups.

  1. It would moreover be relevant to relate the used in vivo dose of PRE (40mg/kg BW) to human consumption.

→ A drug approved by U.S. FDA and most widely used medication for diabetes which activates AMPK, has been treated 850mg~2550mg/day for adults with type 2 diabetes mellitus. Using the FDA guideline to calculate human equivalent does (HED), HED of PRE (effective does in mice: 40mg/kg) is 195mg/day in human. This dose is lower than that of metformin even though PRE is crude extracts.

  1. I am surprised to see that the reviewers did not include a LFD reference group in their in vivo setup to corroborate HFD-induced weight gain. The end BW is surprisingly low for this mouse strain and experimental setup.

è This is a very good point. We used C57BL/6J mouse, commonly used in HFD-induced obese experiment. Also, in many researches related to metabolic disease, HFD-fed mice were mainly used as in vivo model. Thus, we tried to show the main results in HFD-fed mice.

As we described above, we induced obesity in mice with HFD for 10 weeks before starting administration of samples and they were around 41g. In several previous studies in our lab, we observed the body weight of normal chow fed mice (NCD) was around 30-32g when that of HFD fed mice was around 41-43g (about increase in 130%). Although we did not directly compare the body weight between HFD and NCD fed mice at this time, we could assume that the body weight of HFD fed mice we used might be ~130% higher than NCD fed mice.

The reason why the final body weight is relatively low might be the stress by chemical administration and GTT. Especially, fasting during GTT dramatically reduces a lot of body weight (around more than 3-4g).

  1. It would beneficial (read highly encouraged) if the authors would depict their bar plots with individual dots per sample, allowing the reader to visually appreciate data distribution between and within groups). Given the rather low n-size and that the data are unlikely to follow Gaussian distribution, I further suggest to depict all graphs as median ± interquartile range.

→ As reviewer’s comment, all graphs were depicted their bar plots with individual dots per samples in revised manuscript. As reviewer’s comment #1, we have re-analyzed statistics for all data.

  1. As root bark is a food supplement I was surprised to see that the authors chose to administer it intraperitoneally instead of orally. Please provide a sound rationale for this decision – the same goes for the glucose challenge during GTT.

→ We agree with reviewer’s comment. But, bark and roots of B. papyrifera also used as traditional Chinese medicine. Thus, we also considered PRE as a medicine. Furthermore, the components of PRE have been reported to inhibit alpha-glucosidase (1). Alpha-glucosidase inhibitors inhibit the absorption of carbohydrates from the gut and may be used in the treatment of patients with type 2 diabetes or impaired glucose tolerance. To exclude the possibility that oral glucose and PRE administration suppresses glucose absorption from the gut, PRE and glucose was administrated intraperitoneally.

Ref)

  1. Ryu et al., Polyphenols from Broussonetia papyrifera displaying potent alpha-glucosidase inhibition. J Agric Food Chem. (2010) 58:202-208.

  1. Please discuss whether the anti-inflammatory effects observed are adipocyte-specific, as no effect was seen on typical macrophage effector molecules (in vivo). To substantiate any claims to this end, it might be beneficial to run (at least some) of the in vitro assays on macrophage cell lines in addition to the currently used adipocytes.

è This is a very good point. We believe that anti-inflammatory effect of PRE on adipose tissue is not adipocyte-specific effect. In the present study, pro-inflammatory gene expressions and signaling were decreased whereas macrophage marker genes and M2 marker genes were not changed. But it did not mean PRE specifically worked to adipocytes in adipose tissue. In our revised manuscript, PRE also blocked pro-inflammatory responses in Raw264.7 cells, well known as macrophages. Thus, we think PRE improves insulin sensitivity by suppressing inflammation in both macrophages and adipocytes in adipose tissue. We have revised the text in line 191-192, 281 and supplementary results (Figure. S3 and S7).

  1. I was surprised to see that the authors did not pursue the in vivo efficacy of any of the two identified effecter molecules (BF and KJ)?

→ We agree with reviewer’s comment. As reviewer’s comment, it need to examine the in vivo efficacy of BF and KJ. Root bark extract seems to have large number of various components, BF and KJ isolated from root bark (PRE) were contained 0.08 % and 0.08% in PRE, respectively. At this time, the amount of them is not enough to examine in vivo. In future study, we really want to show the physiological roles and in vivo efficacy of two chemicals.

  1. On a similar note, the in vitro concentration of PRE is based on mass (Figure 1 and 2), whereas its derivates are depicted in molar concentrations. Please be consistent and use molar throughout, hence providing the reader a change to look at the potency of the different extracts. To this end, it would also be helpful if the authors could educate the reader on the relative concentration of BF and KJ in the intact PRE.

→ Because PRE contains large number of various components, the concentration of PRE was based on mass. Whereas BF and KJ is a single compound, it was depicted in molar concentration.

  1. Title: The authors cannot state via AMPK without using relevant KO models… As a minimum, they should add ‘potentially via’. To further substantiate their claims, ot would be relevant with AMPK KD cell experiments to show that AMPK signaling is necessary for the immunometabolic effects of PRE and its derivates (BF an d KJ).

→ As reviewer’s comment, we have revised the title. Broussonetia papyrifera root bark and its bioactive compounds suppress adipose tissue inflammation and ameliorate insulin resistance potentially via AMPK activation”

Specific comments:

Introduction:

  1. Line 65-67 needs a bit of grammatical and linguistic help… Unclear what the authors mean. When revising this sentence, please also tone it down to avoid phrases as ‘the strong evidence…’. Suggest to delete ‘strong’ and rephrase to include ‘the observations corroborate’ rather than ‘the evidence’.

è As reviewer’s comment, we have revised the text in line 67. 

  1. Line 76: Delete ‘strongly’.

→ As reviewer’s comment, we have revised the text in line 78.

  1. Line 78: Change ‘tried’ to ‘aimed’ or alternatively delete ‘tried to’.

→ As reviewer’s comment, we have revised the text in line 80.

  1. Line 79: Change ‘the major bioactive…’ to ‘a major bioactive…’

→ As reviewer’s comment, we have revised the text in line 81.

Materials and Methods:

  1. Section 2.4: Did the authors include a mock/vehicle control? It seems so from the result section bu tit is not explicitly mentioned in the M&M. Please amend.

→ As reviewer’s comment, we have revised the text in line 133.

  1. Section 2.5:
  2. How were the animal housed? Singly or grouped?
  3. Overnight fast? Please specify the hours. Was it 12h from e.g. 7PM to 7AM allowing the mice to eat prior fasting or was it from e.g. 4PM to 8AM, which essentially corresponds to 24 hours as mice a nocturnal animals, hence have negligible food-intake throughout the day.
  4. How often did the authors change the food, water etc.? How was the water treated.?
  5. The authors provide the amount of PRE, but not the concentration/volume administered. Please specify.

→ è As reviewer’s comment, we revised the text in section 2.5 Animal experiment.

  1. Section 2.8:
  2. Would suggest to depict individual values (bar plots showing all points) to better appreciate data distribution and allow the reader to instantly assess the number sample number per group for each assay.
  3. Before statistical testing, please assess if the data follows Gaussian distribution and apply the relevant statistical method accordingly.
  4. The authors describe their statistical approach to be students T-test when comparing two groups, but what analysis/post hoc test did they apply to adjust for multiple comparisons when relevant (e.g. Figure 1A-C, 2A, 6C-E)?
  5. It will be important that the authors also correct for multiple testing hypothesis in Figure 4B-C and 5C-D to mitigate the risk of false positive.

→ As reviewer’s comment, we have re-analyzed statistics for all data. Statistical significance was estimated by an unpaired t‐test for comparisons between two conditions. A one‐way ANOVA was used for comparisons between more than two conditions. Dunnett’s post hoc test was used for multiple comparisons. Furthermore, all graph was depicted their bar plots with individual dots per sample in revised manuscript. We have revised the text in the section 2.8. Statistical analysis.

Results:

  1. Please explain what AICAR is and why you have used it as a positive controls (Figure 1C). In its current version and without any reference to the compound, the relevance is not readily accessible for the average reader. It is mentioned in the description of Figure 2 but not 1. Please move the initial description up to Figure 1.

→ As reviewer’s comment, we have revised the text in line 177-178.

  1. Figure 1: The authors mentions that the data (all panels) are representative of three independent experiments, but they do not inform of the actual n-size per experiment. Again, individual dots would help a lot on this matter.

è As reviewer’s comment, all graph was depicted their bar plots with individual dots per sample in revised manuscript.

  1. The concentrations (20µg/ml) seems rather high for in vitro assay. It is 2.000 times higher than the TNF-a doses used…

→ The concentration of TNF-a is generally used in 3T3-L1 adipocyte for activating inflammatory responses (1-3).

Ref)

  1. Suganami T, et al., “A paracrine loop between adipocytes and macrophages aggravates inflammatory changes: role of free fatty acids and tumor necrosis factor alpha,” Arterioscler Thromb Vasc Biol, vol. 25 pp. 2062-2068, 2005
  2. Nepali et al., Luteolin is a bioflavonoid that attenuates adipocyte-derived inflammatory responses via suppression of nuclear factor-κB/mitogen-activated protein kinases pathway. Pharmacogn Mag. 11: 627-635
  3. Gonzales et al., Curcumin and resveratrol inhibit nuclear factor-kappaB-mediated cytokine expression in adipocytes. Nutr Metab (Lond). 5:17

  1. Figure 2C: It would further be appreciated if the PRE concentrations could go from low -> high in panel C as it does in all other panels. Change of orientation does not help on the interpretation.

è To show the inhibitory role of PRE in inflammatory responses more effectively, we aimed to arrange the highest concentration of PRE right next to TNF-a only treated group in Figure 2C.

  1. Figure 3A: I suggest to change the symbols to open circles for the PRE group, allowing the reader to better distinguish between the groups. Also, it is difficult to believe the P values in this panel (especially T120), please double check n-size etc.

→ As reviewer’s comment, we have changed the symbols to open circles for the PRE treated group. And, we also confirm the p value, it is correct.

  1. Figure 3B: Please provide a weekly bodyweight recording, compare it to LFD fed reference mice and monitor the food intake on a daily basis the 7 days the treatment persist as a modestly reduced food intake could explain the observed in vivo findings.

→ This is a very good point. Unfortunately, we did not measure the food intake when we performed animal experiment. However, the significant difference of the body weight between vehicle and PRE treated groups was not observed during experiment. We started experiment when the average body weight was 41.51g and 41.58g for vehicle and PRE treated group, respectively. These groups were weighted 40.03g and 40.11g in previous day of GTT. The average body weight for vehicle and PRE treated group was 35.75g and 35.4g at the end of the experiment, respectively. Thus, we inferred from body weight that there might be no difference for food intake between two groups.

  1. Figure 3 C: It is not clear whether the insulin levels are in fasted or fed state. Please specify.

→ As reviewer’s comment, we have revised in line 222.

  1. I wonder if the authors have insulin data for the entire GTT. Would be beneficial.

è This is a very good point. But we did not collect blood when we performed GTT. To the next time, we will do as reviewer’s comment.

  1. Figure 4B and C: As mentioned above, please adjust for multiple comparisons as all analyses in these panels are related, and so some targets may show up as significantly different merely by chance (false positive).

→ As reviewer’s comment, we first re-analyzed statistics for all data. Statistical significance was estimated by an unpaired t‐test for comparisons between two conditions. A one‐way ANOVA was used for comparisons between more than two conditions. Dunnett’s post hoc test was used for multiple comparisons. But, Fig 4B and 4C are animal experiment with two groups. Thus, we think these data could be analyzed by unpaired t-test for comparison between two conditions. We have revised the text and figures.

  1. Line 231-234: Again, please tone down some of the statements and delete ‘strongly indicate that PRE has strong anti-inflammatory..’ At best, the results indicate that there are some anti-inflammatory activity. Once adjusting for multiple comparisons, a maximum two target genes will come out as significantly decreased.

è As reviewer’s comment, we have revised in text in line 244-245.

  1. Line 251: Delete strongly.

è As reviewer’s comment, we have deleted.

  1. Figure 5A: It would be helpful with an overview picture of the entire liver slide followed by a high resolution (10x or 20x zoom) snap shot. I further suggest to include pictures of all livers in the supplementary file. It is a bit overreaching to conclude diminished steatosis based on a snapshot of a single sample per group. To this end, the manuscript would also benefit from having measured hepatic TG to biochemically support the reported histological findings.

→ As reviewer’s comment, we have added snapshot of liver slide per all animal groups in Figure S6.

  1. Figure 5C and D: Adjust for multiple comparisons.

→ As mentioned above (question #9), Fig. 5C and 5D are also animal experiment and we performed in vivo experiment with two groups. Thus, we think these data could be analyzed by unpaired t-test for comparison between two conditions. We have revised the text and figures.

Discussion:

  1. Line 295: Change ‘In the present study, we tried to focus…’ to ‘In the present study, we focused’.

→ As reviewer’s comment, we have revised the text in line 309.

  1. Line 301: Based on the provided results the authors cannot state that the effects of PRE, BF and KJ are dependent on AMPK as oppose to cooccur with AMPK activation. Please only phrase it as dependent on, if you have formally shown this, either in vitro or in vivo assays, by relevant KD or KO models.

→ Thank you for comment. As reviewer’s comment, assay for KD and KO of AMPK will be further required to explain effects of PRE, BF and KJ are dependent on AMPK. Instead of KD or KO, we presented AMPK dependency of PRE effect by co-treatment with AMPK inhibitor, compound C. Thus, we think that we can explain the effect of PRE is partially AMPK dependent. We have revised the text in the line 315; “partially dependent”.

We highlighted the revisions for reviewer's comments in red and in re-submitted manuscript.

Reviewer 2 Report

The authors of study titled “Broussonetia papyrifera root bark extract exhibits 2 anti-inflammatory effects on adipose tissue and 3 improve insulin sensitivity via AMPK activation” investigated the effects of extracts and bioactive components of Broussonetia papyrifera root bark 18 (PRE) on inflammation and insulin sensitivity. The study was carried out using both in vitro and in vivo animal model. However, before publication manuscript need to be revised. There are some meaningful mistakes and lack of some important information, the most important are listed below:
1. Based on what experiment were chosen extracts and compounds concentration used in this study? Lack of cytotoxicity experiment, we don’t know if the effect of extracts and compounds on cells are not linked with cytotoxic effect!
2. In Materials and Methods 2.3. section poor describing experiment condition. No information on which day of differentiation process experiment was executed, how and when the extract or compound was added to the cell culture?
3. Were this extracts and compounds water soluble?
4. In western blot analysis what software was used to extrapolate the bands on bars? How authors know is the results of western blot analysis (Fig, 2 B and C) were significant? There is no statistical analysis! How authors quantified the protein expression if there is no control protein expression? On Fig. 2B and C lack of control protein(s), this caused the normalized data are unusable for making comparisons!!
5. The same with Fig.4 A and Fig. 5B what software was used to extrapolate the bands on bars? Based on what quantification authors made statistic?
6. Fig. 6 how was chosen BF and KJ concentration for experiment? Were these concentrations nontoxic for 3T3-L1 cells? Fig. 6B and D western blot analysis what software was used to extrapolate the bands on bars? How authors know is the results of western blot analysis were significant? There is no statistical analysis! How authors quantified the protein expression if there is no control protein expression? Lack of control protein(s), this caused the normalized data are unusable for making comparisons!!
7. Authors shoud also post photos of differentiated adipocytes to be sure that the differentiation process was carried out correctly.

Author Response

  1. Based on what experiment were chosen extracts and compounds concentration used in this study? Lack of cytotoxicity experiment, we don’t know if the effect of extracts and compounds on cells are not linked with cytotoxic effect!

→It is a good point. We performed the experiments (nitrogen oxide (NO) measurement and MTT assay) in Raw264.7 cells to show the biological activity and cytotoxic effect of PRE. The activity of PRE is not linked with cytotoxic effect. We have added the results in Figure S3 and described in line 191-192.

  1. In Materials and Methods 2.3. section poor describing experiment condition. No information on which day of differentiation process experiment was executed, how and when the extract or compound was added to the cell culture?

→As reviewer’s comment, we have revised the text in material and methods 2.3 section.

  1. Were this extracts and compounds water soluble?

→Extracts and compounds were dissolved in DMSO. We have revised the methods 2.2 section (line 116-118).

  1. In western blot analysis what software was used to extrapolate the bands on bars? How authors know is the results of western blot analysis (Fig, 2 B and C) were significant? There is no statistical analysis! How authors quantified the protein expression if there is no control protein expression? On Fig. 2B and C lack of control protein(s), this caused the normalized data are unusable for making comparisons!!

→As reviewer’s comment, we have deleted “significant” in text for Fig. 2B and C. Furthermore, we have revised the text in line 196-197. In the present study, phosphorylation of AMPK, ACC, NF-kB a were changed by treatment with PRE while total form of each protein were not changed, respectively. Because the total form of each protein could be the control for phosphorylation, we were able to normalize with total forms.

  1. The same with Fig.4 A and Fig. 5B what software was used to extrapolate the bands on bars? Based on what quantification authors made statistic?

→The images (bands) were subjected to densitometric analyse using ImageJ version 1.49. Statistical comparisons were analyzed by unpaired t-test between two conditions. All statistics were performed with Prism 7.0 (GraphPad Software Inc., San Diego, CA, USA). A p value of < 0.05 was considered to be statistically significant.

  1. Fig. 6 how was chosen BF and KJ concentration for experiment? Were these concentrations nontoxic for 3T3-L1 cells? Fig. 6B and D western blot analysis what software was used to extrapolate the bands on bars? How authors know is the results of western blot analysis were significant? There is no statistical analysis! How authors quantified the protein expression if there is no control protein expression? Lack of control protein(s), this caused the normalized data are unusable for making comparisons!!

→It is a good point. We performed the experiments (nitrogen oxide (NO) measurement and MTT assay) in Raw264.7 cells to choose the most effective concentration of BF and KJ. We have added the results in Figure S7 and described in line 280-281.

In addition, as reviewer’s comment, we have deleted ‘significantly’. It might be a same question about normalization of phosphorylation. The phosphorylation of AMPK, ACC, and NF-kB were changed by treatment with PRE while total form of each protein were not changed, respectively. Because the total form of each protein could be the control for phosphorylation, we were able to normalize with total forms.

  1. Authors shoud also post photos of differentiated adipocytes to be sure that the differentiation process was carried out correctly.

→As reviewer’s comment, we have shown the photo in Figure S2.

We highlighted the revisions for reviewer's comments in red and in re-submitted manuscript.

Round 2

Reviewer 1 Report

I appreciate your effort to improve your manuscript in a timely manner, but is utterly disappointed about the 10-days’ time frame for major revisions provided by the journal; in particular as we are in the midst of a reproducibility-crisis era. While I understand – and agree – on the idea of limiting animal use for experimental research, we also have to understand that some differences occur just by chance with an n-size of 5-6 mice per group. It is therefore good scientific practice to repeat such experiments, and will eventually limit the use of experimental animals, as it will ensure a higher proportion of true findings being published, rather than false negative results that are then pursued by other groups merely to end up in a drawer somewhere, as negative results are immensely difficult to publish.

All that said, the guidelines of the journal can hardly be your responsibility and I will therefore not let this influence my recommendation of your paper.  

General comment for the revised manuscript:

I appreciate your improved statistics as well as providing individual dots per measure instead of just MEAN±SEM. To the same end, I appreciate your inclusion of H&E HPF of all mice in the supplement.

The explanation of dose is relevant and I would thus like you to include this information in the manuscript allowing the reader to better understand the rationale for – and relevance of – the chosen dose.

Please include your argumentation for administering RPE i.p. instead of orally in the discussion.

Author Response

General comment for the revised manuscript:

I appreciate your improved statistics as well as providing individual dots per measure instead of just MEAN±SEM. To the same end, I appreciate your inclusion of H&E HPF of all mice in the supplement.

The explanation of dose is relevant and I would thus like you to include this information in the manuscript allowing the reader to better understand the rationale for – and relevance of – the chosen dose.

→Thank you for your comment. We have revised the text in line 334-336.

Please include your argumentation for administering RPE i.p. instead of orally in the discussion.

→Thank you for your comment. We have revised the text in line 238-241.

Reviewer 2 Report

I still have some concern about this manuscript. Generally section materials and methods are very short and poor in comparison to the result section and all supplementary data. I don’t understand explanation that MTT (cytotoxicity experiment was conducted on Raw cells –there is no such experiment mentioned in Method section, all cells are different and such experiment (MTT) should be prepared on the same cells like all experiments to make any conclusions and comparisons.
Next very important issue is how is describe in Method section 2.5. Animal experiment, in my opinion to this method are result in 3.3. PRE improves obesity-associated systemic glucose tolerance,
But there is more result from experiments on animals like adipose tissue in vivo and on liver (hepatic steatosis), such experiments are not described in method section!!!!! How was conducted are executed such experiments?
Other issue are photos of fully differentiated 3T3-L1 cells (day 7), the cells don’t look like fully differentiated adipocytes, the rate of differentiation is very poor and cells look like there were some cytotoxic effect.

Author Response

I still have some concern about this manuscript. Generally section materials and methods are very short and poor in comparison to the result section and all supplementary data. I don’t understand explanation that MTT (cytotoxicity experiment was conducted on Raw cells –there is no such experiment mentioned in Method section, all cells are different and such experiment (MTT) should be prepared on the same cells like all experiments to make any conclusions and comparisons.

→As the reviewer’s comment, we have added new pictures of preadipocytes and adipocytes treated PRE (40 μg/ml), BF (20 μM) and KJ (20 μM) for 24h in Figure S2B. As you can see, we did not observe any changes of cell morphology or cell death by treating chemicals. The protocol for taking pictures of 3T3L1 adipocytes was added in line 128-132.

In addition, we have newly added Methods section 2.5. NO production and cell viability in Raw 264 cells (line 140-147).

Next very important issue is how is describe in Method section 2.5. Animal experiment, in my opinion to this method are result in 3.3. PRE improves obesity-associated systemic glucose tolerance. But there is more result from experiments on animals like adipose tissue in vivo and on liver (hepatic steatosis), such experiments are not described in method section!!!!! How was conducted are executed such experiments?

→As the reviewer’s comment, we have revised the text in Methods for animal experiments (line 158-160, 161-164, 166, 173-174).

Other issue are photos of fully differentiated 3T3-L1 cells (day 7), the cells don’t look like fully differentiated adipocytes, the rate of differentiation is very poor and cells look like there were some cytotoxic effect.

→We have added new pictures of fully differentiated adipocytes and oil-red O stained adipocytes in Figure S2. In general, 3T3L1 cells were adipogenic differentiated over 80%. Furthermore, we have added new pictures for adipocytes and oil-red O stained adipocytes treated PRE (40 μg/ml), BF (20 μM) and KJ (20 μM) for 24h in Figure S2B. As you can see, we did not observe any changes of cell morphology or differentiation status by treating chemicals. The protocol for oil-red O staining of 3T3L1 adipocytes was added in line 128-132.
